# What's in Phishers: A Longitudinal Study of Security Configurations in Phishing Websites and Kits

## Abstract

Phishing attacks pose a significant threat to Internet users. Understanding the security posture of phishing infrastructure is crucial for developing effective defense strategies, as it helps identify potential weaknesses that attackers might exploit. Despite extensive research, there may still be a gap in fully understanding these security weaknesses. To address this important issue, this paper presents a longitudinal study of security configurations and vulnerabilities in phishing websites and associated kits. We focus on two main areas: (1) analyzing the security configurations of phishing websites and servers, particularly HTTP headers and application-level security, and (2) examining the prevalence and types of vulnerabilities in phishing kits. We analyze data from 906,731 distinct phishing websites collected over 2.5 years, covering HTML headers, client-side resources, and phishing kits. Our findings suggest that phishing websites often employ weak security configurations, with 88.8% of the 13,344 collected phishing kits containing at least one potential vulnerability, and 12.5% containing backdoor vulnerabilities. These vulnerabilities present an opportunity for defenders to shift from passive defense to active disruption of phishing operations. Our research proposes a new approach to leverage weaknesses in phishing infrastructure, allowing defenders to take proactive actions to disable phishing sites earlier and reduce their effectiveness.

## 1 Introduction

Phishing, a type of social engineering attack, has emerged as a major security risk to billions of Internet users [56]. In a typical phishing campaign, attackers craft deceptive phishing websites that masquerade as legitimate websites, such as financial institutions or social media platforms (*e.g.*, PayPal and Facebook), to allure victims and steal sensitive information (*e.g.*, login credentials).

To launch phishing attacks, attackers need to build and host a phishing website. In particular, the building process includes creating client-side (*e.g.*, HTML, JavaScript, and CSS) and server-side resources (*e.g.*, PHP [42]). Attackers can utilize phishing kits that contain pre-configured server-side and client-side resources for phishing websites. Phishing kits have become popular as they significantly ease the development and deployment of phishing websites [7, 13, 26, 51]. Regarding launching phishing websites, attackers may have two options based on how they create them. As the *first* option, attackers also leverage web content publishing services such as website builders (*e.g.*, wix.com [58]) and blogging services (*e.g.*, blogger.com [9]) as they provide a convenient way to create and host web content. As the *second* option, attackers may run standalone web servers running Apache [55], with the pre-configured phishing kits.

Understanding phishing websites is crucial for advancing phishing detection because these sites are central to attackers' efforts to deceive victims and steal sensitive information [38, 39, 41]. As phishing tactics continually evolve, attackers devise new evasion techniques to bypass detection, resulting in a cat-and-mouse game with detection systems. These detection systems often rely on lists of known phishing sites, which, while effective, may not keep up with the latest tactics [33, 59]. To improve detection methods and better protect users, it is essential to deeply understand the phishing ecosystem, including how phishing websites are constructed and maintained. Previous studies have examined various aspects, such as the visual characteristics and use of client-side resources like JavaScript, HTML, and CSS [3, 26, 32, 38, 39, 41], evasion techniques [38, 39, 41], and the role of phishing kits in building these sites [7, 13, 26, 40, 51].

Despite significant research on phishing attacks, the security configurations of phishing websites and servers remain underexplored. To address this gap, we focus on observable artifacts such as (1) identifying vulnerable points in phishing websites by analyzing insecure server configurations, such as missing or improperly set security headers; (2) detecting vulnerabilities within phishing kits that can be exploited; and (3) leveraging these insecure configurations and vulnerabilities to actively exploit phishing kits and disrupt their operations. Analyzing these areas helps us identify common weaknesses that can serve as features for detecting phishing sites. This knowledge also enables us to develop effective tools that not only detect phishing sites but also exploit these identified vulnerabilities to actively disrupt their operations. This approach shifts the paradigm from a purely defensive stance—such as detecting phishing websites and educating users to avoid them—to a proactive one where we actively seek out phishing sites and use their weaknesses to neutralize them [10, 21, 29]. These investigations have contributed to understanding the phishing ecosystem and developing countermeasures against phishing attacks by exploiting a phishing ecosystem.

We conduct a systematic analysis of security configurations in phishing servers and the security implications of phishing kits by collecting HTTP headers, screenshots, client-side resources (*e.g.*, HTML, JavaScript, images), and phishing kits. This data is collected between Jul. 2021 and Jan. 2024 (2 years and 7 months) by accessing 16.7 million distinct phishing URLs and refining the dataset. To identify potential exploit points in phishing websites, we begin by analyzing their security configurations, focusing on HTTP headers (*e.g.*, `Content-Security-Policy` (CSP), `Set-Cookie`) and vulnerabilities in web server program versions. Then, we compare our findings with the security configurations of benign websites (Tranco top 10K websites [52]) to accommodate our analysis.

We observe that security-related HTTP headers are barely employed, or even when used, they are often improperly configured in phishing websites. For example, only 5.4% of phishing websites utilize CSP, while most benign websites (75.2%) use it. Moreover, over 98% of the phishing websites, which specify the `Set-Cookie` header, use the improper directive and its value '`Set-Cookie=/`;'

allowing all directories to access the cookie. Also, 14.3% of the phishing websites are hosted on web servers with known vulnerabilities (*e.g.*, Apache/2.4.6, PHP/7.4.33). These insecure configurations and unpatched systems present further exploitation opportunities, which can be potential avenues for disrupting phishing operations.

Furthermore, we analyze our collected 13,344 distinct phishing kits (2.68M PHP scripts) using two static analysis tools (Semgrep [47] and progpilot [50]), identifying a wide range of security issues, including vulnerabilities that can be immediately exploitable (*e.g.*, backdoors) by security entities. Specifically, we find 689,963 Common Weakness Enumerations (CWEs) from 11,853 phishing kits (88.8% out of 13,344). In particular, we find potentially exploitable security weaknesses, such as CWE-79 (Cross-site scripting) [18] and CWE-89 (SQL injection) [19], are commonly observed in our dataset as well as readily exploitable vulnerabilities (or backdoors) in 1,668 (12.5% out of 13,344) phishing kits.

Our contributions are summarized as follows:

- We conduct a longitudinal analysis of the security configurations of the phishing websites and vulnerabilities within phishing kits by investigating our collected dataset of HTTP headers, client-side resources, and phishing kits for 31 months (Jul. 15, 2021, to Jan. 31, 2024).
- We discover that phishing websites have relatively weak security configurations from their HTTP headers (*e.g.*, CSP and Set-Cookie). In particular, we find that self-hosted phishing websites often misconfigure security-related headers.
- We find 689,963 CWEs (Common Weakness Enumerations) and 1,668 backdoors (or readily exploitable vulnerabilities) in our phishing kit dataset. Additionally, our analysis shows that the phishing kit examined in the case study, which was involved in 204 phishing campaigns, can be easily exploited, suggesting inadequate attention to security measures.
- We publicly share our source code and the collected phishing dataset to facilitate future phishing research upon acceptance.

## 2 Background

Phishing attacks represent a sophisticated social engineering attack where cybercriminals fool victims into disclosing sensitive information. In this section, we introduce how a phishing ecosystem can be configured using HTTP headers and phishing kits.

### 2.1 HTTP Headers and Security Configurations

**HTTP Headers.** HTTP headers transmit additional information between clients and servers during requests and responses, specifying details about the requested resource or desired behaviors. For example, a request header may include the resource's location (*e.g.*, 'GET /index.html') and acceptable data formats ('Accept: text/html'). Similarly, response headers provide details about the server's response, such as status codes ('200 OK'), content types ('Content-Type: application/json;'), and content encoding ('Content-Encoding: gzip'). This information, formatted as key-value pairs, helps ensure accurate data handling by the client. Standard header fields are defined in the IANA registry [37]. **Non-standard Headers.** Non-standard HTTP headers are custom headers not included in the official HTTP/1.1 specification [23] or the IANA registry [37]. Although they were once recommended to be prefixed with 'X-,' this practice was deprecated in 2012 [46].

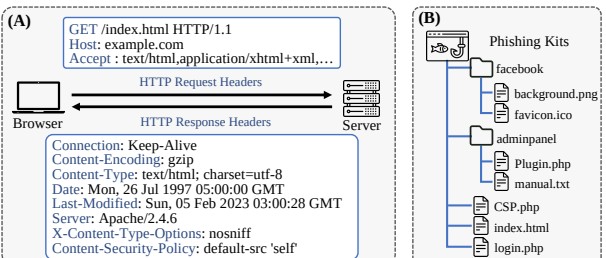

Figure 1: Example of (A) HTTP Header and (B) Phishing Kit.

For instance, modern browsers support the non-standard header X-Content-Type-Options, which instructs clients to strictly follow the MIME types specified in the Content-Type header, preventing MIME type sniffing.

**Security Configuration in HTTP Headers.** HTTP headers help enhance web application security, particularly against Cross-site Scripting (XSS) attacks [14], where attackers inject malicious scripts (*e.g.*, JavaScript) into trusted web pages. For end-users, these scripts appear as legitimate content, leading to the execution of malicious code within the trusted site. Once executed, the scripts can access sensitive information, such as session cookies, potentially hijacking sessions, defacing websites, or redirecting users to harmful sites. To prevent XSS attacks, the CSP header is used to control the sources of permitted content. For example, the directive "Content-Security-Policy: default-src 'self'" restricts content to the local origin, as shown in Figure 1–(A).

**Security Configuration in HTML.** HyperText Markup Language (HTML) is the standard language for structuring web pages using tags, elements, and attributes. It can also help mitigate web security attacks, such as XSS. For instance, a Content-Security-Policy (CSP) can be defined in the HTML '<head>' to restrict scripts to the same origin, using a directive like <meta http-equiv="Content-Security-Policy" content="default-src 'self';">.

### 2.2 Phishing Kit

Phishing kits serve as comprehensive toolsets for deploying phishing sites on web servers [35]. While some creators closely guard their kits, others offer them as part of the cybercrime-as-a-service ecosystem [34]. Specialized criminals develop and sell these kits, often accommodating custom requests [8].

As shown in Figure 1–(B), these kits generally include: (1) the template mimicking the legitimate website's resources (*e.g.*, HTML, images, fonts), (2) pre-compromised web servers (referred to as "shells" or "cpanels") for capturing and transmitting submitted data, and (3) optional features to filter unwanted traffic or implement anti-detection measures. These help significantly lower entry barriers for attackers, enabling individuals with minimal technical expertise to engage in successful phishing operations.

A typical phishing operation involves purchasing a kit, customizing it with a designated email address, uploading and extracting it on a pre-compromised server, and using spam tools to distribute pre-crafted messages to target email lists. The phisher then awaits the influx of stolen credentials.

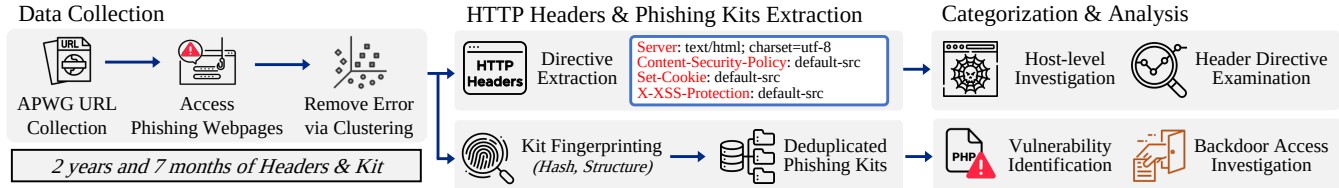

**Figure 2: Overview of Our Systematic Measurement Study.**

## 3 Motivation

The motivation for our research is to identify and exploit weaknesses in phishing infrastructure to actively disrupt and neutralize phishing operations. By analyzing vulnerabilities in phishing websites and kits, we uncover common flaws that can be used not only as indicators for detecting phishing sites but also as entry points for taking direct action against them. This approach moves beyond traditional defensive strategies—such as detecting phishing sites and raising user awareness—toward a more proactive stance. Instead of merely reacting to phishing threats, we aim to leverage the attackers' own weaknesses to interfere with their operations, effectively turning the tables and neutralizing the threat at its source. This shift in strategy aims to significantly reduce the impact of phishing attacks by disrupting them before they can exploit victims.

## 4 Crawler Design & Dataset Collection

As shown in Figure 2, our systematic measurement study comprises three phases: data collection, HTTP headers and phishing kits extraction, and categorization and analysis. We designed a web crawler that systematically accesses real-world phishing websites, gathering APWG URLs and removing errors via clustering. The crawler collects HTTP response headers, client-side resources (*e.g.*, HTML), screenshots, and phishing kits. Deployed from July 2021 to January 2024 (931 days), it accessed 16,742,415 (16.7M) phishing websites. The extraction phase focuses on HTTP headers (*e.g.*, CSP) and kit fingerprinting. The analysis phase involves host-level investigation, header examination, vulnerability identification (focusing on PHP), and backdoor access investigation. This approach enables in-depth analysis of phishing techniques and infrastructure.

### 4.1 Phishing Crawler Design

**Phishing Website Resource Crawler Design.** We design a web crawler that periodically (every 10 minutes) collects phishing website resources such as images, DOM, and HTTP response headers. Our crawler also captures screenshots of the phishing websites after fully loading and executing client-side resources. These screenshots help validate the authenticity of reported phishing URLs and detect potential access errors. We implement the crawler using Google Selenium ChromeDriver [12], which simulates real user interactions with phishing websites. This approach provides a comprehensive view of the phishing webpages by fully rendering all client-side resources and helps evade anti-bot techniques that might otherwise block our crawler [5, 31].

**Phishing Kit Crawler Design.** In addition to collecting phishing websites' client-side resources (*e.g.*, HTML) and HTTP header information, our approach also focuses on gathering phishing kits actively used in real-world attacks. However, identifying these kits

**Table 1: Overview of Our Collected Dataset from July 2021 to January 2024 (31 months).**

| Type | # of URLs | # of Websites* |
|---|---|---|
| Total APWG Phishing Reports | 16,742,415 | 1,845,523 |
| Successfully Accessed URLs | 7,807,532 | 1,466,343 |
| Screenshots | 6,832,416 | 1,221,807 |
| Final Refined Dataset | 3,543,349 | 906,731 |

**# of Clusters from Refined** = 544,173;
**# of Total Kits** = 18,865; **# of Refined Kits** = 13,344;
∗ Distinct phishing websites;

poses a significant challenge due to the lack of specific information about their locations (*i.e.*, paths) and filenames on phishing web servers. To address this issue, our approach leverages the observation that phishing attackers may leave their kits publicly accessible and downloadable at certain URL paths, even after deploying the phishing websites using these kits [13, 51].

The attackers' oversight enables our crawler to discover and download phishing kits from publicly accessible locations on compromised web servers. Specifically, the crawler visits each phishing URL and checks if directory listing (or directory indexing) is enabled. If it is, the crawler initiates a recursive process to download all files, including compressed archives (*e.g.*, zip, rar, tar, 7z) and other resources such as images, HTML, and JavaScript within the directory structure. When directory listing is not enabled, or no files are found in the initial directory, the crawler systematically navigates to the parent directory of the phishing URL in an attempt to locate accessible files.

### 4.2 Phishing Data Collection

**Collecting Phishing Website Resources.** Our crawler is fed real-world phishing URLs from the APWG eCrime Exchange (`eCX`) in real time [6]. `eCX` is one of the largest phishing attack report repositories containing reliable real-world phishing attack reports. This repository has been widely used to better understand the phishing attack ecosystem [28, 38–40, 60]. As illustrated in Figure 2, our crawler runs every 10 minutes from July 15th, 2021 to January 31st, 2024 (2 years and 7 months). During the collection period, our crawler is fed a total of 16,742,415 (16.7M) real-world phishing URLs from APWG `eCX`. Out of 16.7M phishing URLs, only 46.6% (7.81M) are accessible; the others are inaccessible due to network errors (*e.g.*, DNS), web servers being offline, etc.

**Refining Collected Phishing Websites Dataset.** We leverage screenshots to identify and filter out internal errors from the dataset. These errors could bias our analysis since error pages may have HTML content and HTTP headers different from those of the original phishing sites. To achieve this, we apply a conservative filtering approach, removing clusters that contain error pages, such as empty

screens, client errors (*e.g.*, error codes 400–499), or server errors (*e.g.*, error codes 500–599). Our dataset initially included 6.8M screenshots from 7.8M accessible phishing websites, with 975K (12.8%) missing due to redirections or ChromeDriver issues.

To efficiently eliminate error pages without manually reviewing all 6.8M screenshots, we employ `Fastdup` [4], an unsupervised tool for analyzing image datasets. This tool helps identify duplicates, outliers, and related images by clustering them. We generate 569,994 clusters, enabling us to target groups containing error messages. For validation, two security researchers *manually* reviewed 23,579 clusters (those with more than seven screenshots), covering over 90% of the dataset, to ensure accurate filtering. After filtering, we retain 544,173 clusters, representing 906,731 distinct phishing websites, which are used for our analysis, as shown in Table 1.

**Refining Collected Phishing Kit Dataset.** Our phishing kit crawler is also fed real-world phishing URLs from the `eCX` platform, visiting each URL every 10 minutes during the same collection period as our phishing website resource collection. We collect 18,865 compressed (*e.g.*, `.zip`, `.7z`, `.rar`) or archived files (*e.g.*, `.tar`) from 1.2M distinct phishing websites.

To prevent duplicate phishing kits from skewing our analysis, we implement a two-step process to ensure dataset uniqueness. *First*, we calculate cryptographic hash values for each kit using the Blake2 algorithm to identify duplicates. *Second*, we compare the directory structures and script file names (*e.g.*, PHP, JavaScript, CSS, images) within each kit as illustrated in Figure 1–(B). If both the hash and structure match, we classify the kits as duplicates and remove them. This approach, used in prior works [7, 51], validates the uniqueness of the phishing kit samples in our collection.

We identify and remove 5,521 duplicate phishing kits (29.3% of 18,865) from our initial collection, averaging 4.4 duplicates per kit ($\sigma$ = 6.28), with a maximum of 97 duplicates. Our final dataset includes 13,344 unique phishing kits, significantly larger than prior work (more than 160 times), which analyzed only 70 kits [7]. These distinct kits are used for security analysis with two static analyzers, `Semgrep` [47] and `progpilot` [50].

## 5 HTTP Response Headers

We conduct a comprehensive analysis of the HTTP response headers of our collected phishing websites. Specifically, we first analyze the overall landscape of phishing website headers, with a particular emphasis on 11 commonly used security-related headers. Then, we conduct an in-depth analysis of headers related to Cross-site mitigation headers, server information contains vulnerabilities using Common Vulnerabilities and Exposures (CVE) information, and misconfiguration.

**Categorization of Hosting Phishing Websites.** Recall that there are two different ways phishing websites are managed from the web server's perspective, *i.e.*, using web content publishing service or self-hosting, which may affect how the servers are configured substantially. Specifically, if using *web content publishing services*, phishing attackers use the pre-built website templates, functionalities, and often their hosting facilities offered by these services. In other words, attackers do not have access to the underlying web server's configurations. On the other hand, if they choose the *self-hosting* option, attackers may have more control over the web server, including

the ability to configure various settings, including HTTP headers. Therefore, our analysis focuses merely on *self-hosting* option.

As the two types have substantially different controllabilities, we categorize our data according to the types. Specifically, we identify which group each website belongs to using the Public Suffix List [43], a resource that identifies common domain suffixes associated with website creation platforms (*e.g.*, wix.com, github.io). Essentially, websites using these suffixes in their domain names likely rely on content publishing services, while others fall into the self-hosting category. To this end, we identify that 38.4% of the phishing websites in our dataset rely on (1) content publishing services and 61.6% of the phishing websites serve on (2) self-hosting web servers, as detailed in Table 6.

**Security-related Header Types.** We first identify security-related headers and categorize them into two groups: those that can actively prevent security vulnerabilities and those that can introduce vulnerabilities if misconfigured. As shown in Table 2, eight headers (highlighted in orange) fall into the first category, while the remaining four headers (highlighted in cyan) belong to the second category. The 'Security' column of Table 2 describes the types of attacks, vulnerabilities, or mitigation associated with each header.

**Collecting Benign Website Resources.** To further compare the security configurations of phishing websites, we collect benign data from the top 10K domains on the Tranco 1M list [52], gathered on August 28, 2024. Using the same crawler employed for phishing websites, we collect resources from these benign domains, including HTTP header information.

### 5.1 Overview of HTTP Header Usage

**General Usage of HTTP Headers.** The top three headers used by phishing websites are `Content-Type` (99.9%), `Date` (99.9%), and `Server` (94.6%), similar to benign sites. Other common headers include `Content-Encoding` (86.5%), `Transfer-Encoding` (77.8%), and `Connection` (69.9%). Security-related headers are much less common, with `Set-Cookie` at 35.5%, `X-Content-Type-Options` at 33.2%, and `Strict-Transport-Security` at 11.4%. `Referrer-Policy` is used the least, at just 3.1%. Compared to benign websites, the general usage of headers is similar (within a 0.1 margin), except for the `Server` header. In benign websites, only 78.4% use the `Server` header, likely to reduce the risk of information leakage and prevent potential vulnerabilities

**Content Publishing Service vs. Self-hosting Server.** Phishing websites on content publishing services use more security-related headers than self-hosted sites, such as `X-Content-Type-Options` (67.5% vs. 12.0%) and `X-XSS-Protection` (65.2% vs. 9.8%). However, `Set-Cookie` is more common on self-hosted servers (53.4% vs. 6.5%).

**Adoption Trend of Security-related Headers.** Figure 3 shows the overall trend of phishing websites and the usage of security-related headers. Headers like `Set-Cookie`, `X-Powered-By`, and `X-Frame-Options` have risen significantly, while others, such as `Content-Type`, `Server`, and `Strict-Transport-Security`, have slight increases. `Expect-CT` declined after September 2022, while `X-Content-Type-Options`, `X-XSS-Protection`, and `Content-Security-Policy` remained steady.

**Comparison with Benign Domain.** Previous studies [25, 30, 45, 54] highlight the growing adoption of security headers such

Table 2: Top 12 Security-related HTTP Headers Used in Content Publishing Service and Self-hosting.

| Security-related Header | Total Usage | | Content Publishing Service | | Self-hosting | | Security Issues |
|---|---|---|---|---|---|---|---|
| | Value | Usage (%) | Value | Usage (%) | Value | Usage (%) | |
| **Content-Type** | utf-8* | 788,996 (87.1%) | utf-8* | 323,495 (93.5%) | utf-8* | 465,501 (83.1%) | A resource might be read |
| | text/html | 112,912 (12.5%) | text/html | 22,101 (6.4%) | text/html | 90,811 (16.2%) | as HTML, creating potential |
| | iso-8859-1* | 616 (0.1%) | iso-8859-1* | 35 (0.1%) | iso-8859-1* | 581 (0.1%) | for XSS vulnerabilities |
| **Server** | Cloudflare | 227,984 (26.6%) | GSE | 220,319 (69.8%) | Cloudflare | 201,063 (37.1%) | |
| | GSE | 220,615 (25.7%) | Cloudflare | 26,921 (8.5%) | Apache | 161,709 (29.8%) | Information leak |
| | Apache | 184,004 (21.5%) | Apache | 19,624 (6.2%) | Nginx | 78,532 (14.5%) | |
| **Set-Cookie** | Path | 318,337 (98.9%) | Path | 22,330 (98.5%) | Path | 296,047 (98.9%) | |
| | HTTPOnly | 199,334 (61.9%) | HTTPOnly | 13,358 (58.9%) | HTTPOnly | 185,976 (62.1%) | Cookie transmission |
| | Secure | 162,386 (50.4%) | Secure | 10,764 (47.5%) | Secure | 151,622 (50.7%) | |
| **X-Content-Type-Options** | nosniff | 300,064 (99.7%) | nosniff | 233,664 (99.9%) | nosniff | 66,420 (98.7%) | Malicious content can be sent |
| **X-XSS-Protection** | 1;mode=block | 272,931 (97.2%) | 1;mode=block | 223,846 (99.2%) | 1;mode=block | 49,085 (89.2%) | |
| | 0 | 5,836 (2.1%) | 0 | 1,781 (0.8%) | 0 | 4,055 (7.4%) | XSS attacks |
| | 1 | 918 (0.4%) | 1 | 56 (0.1%) | 1 | 862 (1.6%) | |
| **X-Powered-By** | PHP | 111,864 (86.1%) | PHP | 7,028 (70.8%) | PHP | 104,836 (87.3%) | |
| | Plesklin | 9,898 (7.6%) | Plesklin | 2,298 (23.1%) | Plesklin | 7,600 (6.3%) | Information leak |
| | ASP.NET | 6,961 (5.4%) | express | 1,451 (14.6%) | ASP.NET | 6,265 (5.2%) | |
| **Strict-Transport-Security** | max-age | 102,720 (99.0%) | max-age | 55,009 (99.9%) | max-age | 47,711 (97.9%) | Can prevent HTTPS |
| | includeSubDomains | 52,435 (50.6%) | includeSubDomains | 29,864 (54.1%) | includeSubDomains | 22,571 (46.3%) | downgrade attacks |
| | preload | 36,249 (34.9%) | preload | 26,906 (48.9%) | preload | 9,343 (19.2%) | |
| **Expect-CT†** | max-age | 51,452 (99.9%) | max-age | 41,031 (99.9%) | max-age | 41,029 (99.9%) | Can prevent the use of |
| | report-uri | 51,249 (99.6%) | report-uri | 10,408 (99.8%) | report-uri | 40,841 (99.5%) | misissued certificates |
| | enforce | 170 (0.3%) | enforce | 50 (0.5%) | enforce | 120 (0.3%) | |
| **X-Frame-Options** | sameorigin | 41,164 (80.3%) | sameorigin | 4,083 (70.9%) | sameorigin | 37,081 (81.5%) | |
| | deny | 6,972 (25.7%) | deny | 1,543 (26.9%) | deny | 5,430 (11.9%) | Can prevent iframe attacks |
| | allowall | 2,302 (21.5%) | allowall | 107 (1.9%) | allowall | 2,195 (4.8%) | |
| **Content-Security-Policy** | upgrade‡ | 23,736 (48.3%) | upgrade‡ | 11,841 (55.4%) | upgrade‡ | 11,895 (55.6%) | Prevent cross-site-scripting |
| | script-src | 18,562 (37.8%) | script-src | 7,590 (35.5%) | script-src | 10,972 (51.3%) | attacks |
| | frame-ancestors | 12,046 (24.5%) | report-uri | 6,523 (30.5%) | default-src | 9,125 (42.7%) | |
| **Access-Control-Allow-Origin** | char (∗) | 42,893 (87.4%) | char (∗) | 21,105 (99.9%) | char (∗) | 21,788 (78.0%) | Extends cross-origin |
| | <origin> | 6,064 (12.4%) | <origin> | 14 (0.1%) | <origin> | 6,050 (21.7%) | resource sharing |
| | null | 30 (0.1%) | null | 2 (0.1%) | null | 28 (0.1%) | |
| **Referrer-Policy** | strict-origin§ | 9,267 (54.1%) | strict-origin§ | 4,726 (84.8%) | strict-origin§ | 4,541 (39.3%) | Restrict the exposed |
| | when-downgrade‖ | 3,493 (20.4%) | same-origin | 315 (5.7%) | when-downgrade‖ | 3,424 (29.7%) | referrer information |
| | same-origin | 2,003 (11.7%) | unsafe-url | 274 (4.9%) | same-origin | 1,688 (14.6%) | |

Orange-colored headers that can directly mitigate attacks; Cyan-colored headers can introduce vulnerabilities if misconfigured; ∗: 'text/html charset=html' is also included;
†: This header is deprecated due to the lack of support by browsers; ‡: upgrade-insecure-requests; §: strict-origin-when-cross-origin; ‖: no-referrer-when-downgrade;

as CSP and Set-Cookie to enhance website security in benign websites. To compare trends, we analyze the top 10K domains from the Tranco 1M list [52]. Our findings show that benign websites have higher adoption rates of security headers compared to phishing websites: Set-Cookie (67.5% vs. 35.5%), CSP (75.2% vs. 5.4%), X-XSS-Protection (75.8% vs. 31.0%), and Strict-Transport-Security (15.9% vs. 11.4%).

> **Takeaway:** Phishing websites often lack proper security headers, with content publishing services generally implementing more safeguards than self-hosted sites. However, self-hosted servers use Set-Cookie more frequently (53.4% vs. 6.5%), implying more persistent user tracking or malicious monitoring of victim behavior. While the use of some security headers has increased, overall adoption remains low. These patterns not only expose vulnerabilities but also provide valuable indicators for detecting and analyzing phishing infrastructure, aiding in the development of more effective anti-phishing strategies.

## 5.2 Cross-site Mitigation Headers

HTTP headers play a critical role in preventing cross-site attacks such as XSS and CSRF. This analysis focuses on two key headers, CSP and Set-Cookie, highlighting significant gaps between security best practices and their implementation in phishing websites.

**Content Security Policy (CSP).** Although CSP has been a standard since 2010, its adoption among phishing websites remains surprisingly low, with only 5.4% of sites implementing it. Content publishing services show slightly higher adoption rates (6.2%) compared to self-hosted servers (4.9%). The upgrade-insecure-requests directive is found in only 2.6% of sites, while the frame-ancestors directive, which is critical for framing control, appears in just 1.3%. Additionally, 74.4% of domains use the <meta> tag for setting HTTP headers, but just 0.8% of those set the CSP header via <meta>.

Most phishing sites are misconfigured, likely because they do not require robust security. This may result from outdated phishing kits or automated tools that have not integrated modern security standards. Among self-hosted phishing sites, 98.3% still specify unsafe-inline in the script-src directive, effectively negating the intended protections of CSP. Additionally, 21.1% misuse the default-src directive. Reporting mechanisms, crucial for detecting violations, are also neglected, with only 3,439 sites using report-uri and just 93 using report-to, none of which are correctly configured.

Regarding TLS enforcement, 48.3% of phishing sites implement upgrade-insecure-requests, but only 9.7% of self-hosted websites combine it with the recommended Strict-Transport-Security header. Additionally, only 46.3% of the applications include subdomains to secure their subdomains.

For framing control, our data indicates a preference for the outdated `X-Frame-Options` header over CSP's frame-ancestors directive. Furthermore, 21.5% of websites misuse the invalid allowall directive, and 93.5% misconfigure the `allow-from` directive.

**Set-Cookie Header.** The `Set-Cookie` header is more commonly used in phishing websites, but often with broad settings. Over 98% of these sites set the `Path` directive to '/,' making cookies accessible across all directories. Only 61.9% use the `HttpOnly` attribute and 50.4% implement the Secure directive.

The `Set-Cookie` header also includes the `SameSite directive`, crucial for preventing CSRF attacks. However, its adoption is particularly low, with only 11.8% of self-hosted sites and 16.9% of all sites implementing it.

TLS enforcement via the `Secure` directive shows a somewhat higher adoption rate, with 50.4% of websites using it alongside `Set-Cookie` headers. This pattern mirrors the emphasis on HTTPS adoption among phishing sites. However, with 37.9% of sites failing to implement `HttpOnly`, a significant proportion remain vulnerable to XSS attacks.

> **Takeaway:** Phishing websites demonstrate limited and often misconfigured use of security headers. CSP implementation is rare and frequently ineffective due to misconfigurations. While the adoption of `Set-Cookie` headers is more widespread, critical security directives are omitted. The higher rate of TLS adoption likely serves to enhance the perceived legitimacy of phishing sites. These distinct patterns in security header usage can serve as indicators for detecting and analyzing phishing infrastructure.

## 5.3 Vulnerabilities in Phishing Websites

The `Server` header frequently discloses information about web server software and its versions, potentially exposing vulnerabilities. Our data shows that phishing websites are more than twice as likely to expose version information compared to benign domains (14.3% vs 6.5%). This increased exposure significantly expands the attack surface of phishing sites. RFC 2616 [44] explicitly states that revealing server details poses a security risk.

**Vulnerable Server Version Exposure.** `Apache` versions `2.4.6` and `2.4.41` contain 21 and 23 known vulnerabilities, respectively. Critical vulnerabilities, such as `CVE-2023-25690` [2] (severity score 9.8), affect `Apache` versions `2.4.0` to `2.4.55`. `Nginx` also has 75 identified vulnerabilities, with versions `1.18.0` and `1.19.10` being the most common. Even displaying server names alone poses a risk, as most `Apache` versions have publicly reported vulnerabilities.

**Server Fingerprinting through Headers.** The `X-Powered-By` header further exposes server-side technologies. PHP is most common, with outdated versions like 5.4.16 (containing 42 known vulnerabilities) widely used. This raises concerns about the potential secondary exploitation of phishing sites.

**Cloud and Hosting Services Usage.** `Cloudflare` and `GSE` (Google's servers) are widely used, with `GSE` serving Blogger.com, accounting for 99.9% of its usage. `Cloudflare`'s services complicate detection, as our crawler cannot identify additional hosting layers.

> **Takeaway:** Phishing websites often expose server information through `Server` and `X-Powered-By` headers (14.3% vs 6.5% in

**Table 3: Top 10 CWEs in Phishing Kits.**

| ID | Type | LoE* | Severity | Semgrep # Vuln. (%) | Progpilot # Vuln. (%) |
|---|---|---|---|---|---|
| CWE-79 | XSS | M | M | 507,244 (73.71%) | 101,589 (38.56%) |
| CWE-89 | SQL Injection | H | M | 88,426 (12.85%) | 43,183 (16.38%) |
| CWE-95, 918 | SSRF | L | H | 49,755 (7.23%) | 5,398 (2.05%) |
| CWE-23 | Path Traversal | H | M | 22,254 (3.27%) | 16,932 (6.42%) |
| CWE-295, 319 | Cleartext Trans. | L | M | 9,484 (1.38%) | 14,827 (5.63%) |
| CWE-98, 489 | Leftover Debug | L | L | 3,265 (0.47%) | 12,209 (4.63%) |
| CWE-94, 601 | Code Injection | L | H | 2,244 (0.33%) | 23,221 (8.81%) |
| CWE-470, 1333 | Unsafe Redirect | M | M | 1,544 (0.22%) | 207 (0.08%) |
| CWE-78 | OS Command Inj. | L | H | 1,286 (0.19%) | 1,035 (0.39%) |
| CWE-614, 1004 | Cookies Set | L | L | 1,245 (0.18%) | 45,319 (17.19%) |

∗ **LoE** = Likelihood of Exploit; **H** = High; **M** = Medium; **L** = Low;
† Ordered by `Semgrep`'s number of Vulnerabilities

benign sites), revealing software versions with known vulnerabilities in `Apache`, `Nginx`, and PHP. Cybersecurity professionals can leverage this information to enhance phishing detection, develop mitigation strategies, and potentially infiltrate phishing operations for intelligence gathering. These vulnerabilities present opportunities for counter-phishing efforts.

## 5.4 Misconfiguration

**Using Ineffective Directives.** Headers enhance functionality but can pose security risks if misconfigured. For example, setting `X-XSS-Protection` to "0" disables XSS protection, yet 2.1% of websites in our dataset still use it. Similarly, 87.4% of websites use the wildcard (*) in `Access-Control-Allow-Origin`, allowing any origin access, with nearly all phishing sites (99.9%) on content publishing services doing so. Additionally, 11.7% of websites use the risky `unsafe-url` value in `Referrer-Policy`, exposing them to data leaks and security vulnerabilities.

**Vulnerability of Non-standard Header.** The `X-XSS-Protection` header, though intended to enable browser XSS filtering, can introduce XSS vulnerabilities under certain configurations. Despite its risks, usage has stabilized at 31.0% across phishing sites, with a particularly high prevalence (65.2%) on Content Phishing Services.

> **Takeaway:** A misconfigured `X-XSS-Protection` header can introduce vulnerabilities, while common configurations in `Access-Control-Allow-Origin` expose websites to unauthorized access and data leaks.

## 6 Vulnerabilities in Phishing Kits

In this section, we look at CWE vulnerabilities and backdoors to code in phishing kits. We conducted penetration testing to scrutinize the level of vulnerability of phishing kits in a local environment, adhering to ethical considerations.

## 6.1 CWE in Phishing Kits

**Overview of Vulnerabilities.** 13,344 distinct phishing kits include 2,685,201 PHP scripts, each containing 205.93 PHP files on average. We run the two static analysis tools, `Semgrep` [47] and `progpilot` [50], on all the PHP scripts we collect. The result shows that phishing kits contain a wide range of vulnerabilities, which are categorized under the Common Weakness Enumeration (CWE) system. Specifically, out of 13,344 kits, 11,853 kits (88.8%) have more than one CWE reported. The result essentially shows that phishing kits might be vulnerable to various attacks.

As shown in Table 3, the top three CWE categories identified by `Semgrep` are CWE-79 [18] (XSS, 73.71%), CWE-89 [19] (SQL Injection, 12.85%), and CWE-918 [20] (SSRF, 7.23%), accounting for 93.79% of the total CWEs detected. On the other hand, the top three CWE categories identified by `progpilot` are CWE-79 (XSS, 38.56%), CWE-1004 [15] (Insecure Cookie, 17.19%), and CWE-89 (SQL Injection, 16.38%), representing 72.13% of the total CWEs reported.

**Severity of CWEs.** To better understand the severity of the CWEs we found, we leverage semgrep's `Likelihood` metric [48], which aims to reflect the impact and ramification of the CWE and focus on the cases with high severity scores. Specifically, CWE-89 (SQL injection) and CWE-502 [17] (deserialization of untrusted data) are the two highly severe CWEs. CWE-79 (XSS) and CWE-22 [16] (path traversal) are of medium and low severity, respectively. While they are not highly severe, they can still lead to significant security breaches. Note that the weaknesses of phishing kits are likely to be reflected in the phishing websites they create.

*6.1.1 Case Study of XSS Vulnerabilities.* Listing 1 shows a code snippet from the `darkx.zip` This shows a code snippet from the phishing kit. It was found to be part of a family of duplicate phishing kits found on at least 70 hosted phishing campaigns. When unprocessed input from HTTP parameters on the login page is passed to an echo statement, the statement renders an HTML page and returns it to the user. It contains XSS vulnerability in the `verify.php` page (lines 1-5), part of a fake Microsoft sign-in UI designed to steal user credentials. When the sign-in form is submitted, the user's email and password are passed directly into a redirect URL without proper validation or encoding. The unsanitized `echo` (line 3) in `verify.php` is the entry point for the XSS payload. By crafting a malicious URL with JavaScript injected into the `email` parameter (line 4), a malicious code can be executed in the victim's browser when the page is loaded. Then, the stolen credentials are sent to the `next.php` page (lines 6-19), where they are logged and exfiltrated. The `next.php` script retrieves the compromised email and password from the `GET` parameters (lines 7-8) and logs them for the attacker (line 11).

Note that if the `verify` parameter is 0, the script redirects the user back to `verify.php` with the `email` from the `GET` request and `error=true&verify=1` (lines 13-15). This redirection uses `window.location` without proper encoding, allowing a further XSS attack. If `verify` is 1, the script starts a new session and redirects to the legitimate Microsoft login page using a meta refresh tag, likely to avoid detection (lines 16-19). Observe that the code lacks input validation and output encoding, making the websites they created vulnerable to various attacks. For example, an attacker can manipulate the parameters of the `verification` and `email` to perform additional attacks or bypass checks in the phishing flow.

*6.1.2 Case Study of SQL Injection Vulnerabilities.* Listing 2 shows an SQL injection vulnerability that we manually verify. Specifically, in the `esestandard.zip` phishing kit, the `$_POST['username']` and `$_POST['password']` are directly concatenated into the SQL query without proper sanitization (lines 2-3). A maliciously crafted input containing malicious SQL commands can be injected and executed as part of the query. This allows the attacker to modify the query's logic, bypass authentication, or extract sensitive data from the database. In addition, the use of the deprecated functions

```php
<?php                                          verify.php
if (isset($_POST['signin'])) {
    echo "<script>window.location='next.php?
        email=".$_GET['email']."&password=".$_POST['password']
        ."&verify=0'</script>";} ?>
<?php                                          next.php
$message .= "Email: ".$_GET['email']."\n";
$message .= "Password: ".$_GET['password']."\n";
// ... collect location info and user agent ...
$fp = fopen('.error.htm', 'a');
fwrite($fp, "\n".$message); fclose($fp);
// ... send email to attacker ...
if ($_GET['verify'] == 0) {
    echo "<script>window.location='verify.php?
        email=".$_GET['email']."&error=true&verify=1'</script>";
} elseif ($_GET['verify'] == 1) {
    session_start();
    echo "<meta http-equiv='refresh' content='0;
        url=https://login.live.com/'/>"; } ?>
```

**Listing 1: Two Examples of XSS Vulnerability.**

```php
<?php require_once('Connections/conn.php');
$user = $_POST['username'];
$pass = $_POST['password'];
$result = mysql_query("SELECT * FROM login WHERE
username='$user' AND password='$pass'") or die(mysql_error());
$row = mysql_fetch_array( $result );
// ... rest of the code ...; ?>
```

**Listing 2: Example of SQL Injection Vulnerability.**

(*e.g.*, `mysql_fetch_array()`) further increases the risk (lines 4-6). These codes are most often found in the code where the phisher sends information to the server and on the redirect page when the victim clicks the website's login button.

> **Takeaway:** We find a number of XSS and SQL injection weaknesses in the phishing kits, suggesting that phishing websites created by them might also be vulnerable. We manually verified the vulnerabilities in the phishing kits and presented them in the case study, demonstrating that these weaknesses can actively disrupt phishing attackers and neutralize them.

## 6.2 Backdoor in Phishing Kits

We further analyze the phishing kits to check whether there are potential backdoors (*i.e.*, intentional vulnerabilities) that adversaries can exploit (against the phishing websites). Specifically, we focus on files containing XSS vulnerabilities, which can be exploited to allow unauthorized access. We use `shellray` [49] and `VirusShare` [53] to identify known patterns of malware and web shells using keywords such as `shell_exec` and `pcntl`. The analysis shows 1,668 phishing kits (12.5% out of 13,344 kits) contain web shell backdoors and 135 known malware. The keywords used in this analysis are listed in Table 5. To scrutinize, we activated and penetration-tested the phishing kit in our controlled local environment, adhering to ethical considerations.

Figure 4(a) shows a backdoor we found in the phishing kit called `pki-validation.zip`, which is found in 204 hosted phishing campaigns. It uses a multi-step execution process to maintain persistent access to phishing sites. When we deobfuscate the script, it determines the execution context using `isCli()`. If it is not running in CLI mode, it bypasses potential restrictions by identifying PHP functions that can be used to execute system commands. Then, it attempts to access the '/robots' URL using `curlRobots()` to evaluate the server configuration. If successful, `runCmd()` executes the

 

command in the background (via `shell_exec()` or `passthru()`), creating a persistent backdoor.

In addition, the backdoor is stealthy. Once the backdoor is established, the script deletes itself using `unlink()` to cover its tracks. It then calls `lockFile()`, which enters an infinite loop to monitor and modify sensitive files. The `lockFile()` checks for the existence of files like '.htaccess,' 'robots.txt,' and 'sitemap.xml' using `get_contents()`, and then calculates the SHA1 hash of each file using `hash()`. If changes are detected, the script uses `create-File()` to overwrite the files with attacker-controlled content. By monitoring and modifying these critical files, the backdoor can be persistent. Note that it can cause "the theft chain of victims' information" where other attackers exploit the backdoors and steal the stolen victims' information from the phishing attackers. However, as shown in Figure 4(b), this backdoor control panel does not have proper traversal settings and can be easily accessed via subdirectory recursion (*e.g.*, ffuf [1]) with the wordlist. In the case of an actual phishing campaign, the control panel may also be able to look up information about that server. We note that those phishing kits are vulnerable. Moreover, law enforcement may take immediate action to disrupt phishing attacks by exploiting the backdoors.

> **Takeaway:** We find that 12.5% of phishing kits contain readily exploitable vulnerabilities, which might be abused by other adversaries who are aware of their existence. We speculate that both defenders and offenders can also leverage them.

## 7 Discussion

**Limitations.** Identifying content publishing services and server owners is difficult when phishing attackers use redirects. To address this, we manually verified 100 phishing websites from content publishing services, comparing provider names from domains and HTML files, and confirming full consistency in the results.

**Recommendations.**

- *Detection Improvement with Fingerprinting*. Improving phishing detection through fingerprinting misconfigured headers like CSP or `Set-Cookie` can enhance accuracy beyond current methods based on visual similarities and suspicious URLs. This approach increases detection reliability and reduces false positives, strengthening defenses against phishing attacks.

- *Understand Taken Information*. Identifying and exploiting security vulnerabilities on phishing servers, such as insecure header configurations or flaws within phishing kits, can provide critical insights into phishing attackers' operations. Accessing the servers or databases of confirmed phishing websites allows attackers to analyze the types of information collected. This analysis enables the development of enhanced defense mechanisms targeted at the most compromised data categories (*i.e.*, the geo-locations, IP addresses, credit card details, and login credentials). Under legal and ethical guidelines, retrieving crucial data can help categorize the victims' profiles and understand the scope of the phishing attack.

**Ethics.** Our methods prioritize ethical considerations while maintaining scientific rigor in the analysis of real-world phishing websites and kits. We have implemented strict protocols to ensure that no victims' information was collected, compliance with the General Data Protection Regulation (GDPR). Moreover, all experimental analyses were conducted within a controlled virtual environment.

This approach allows us to examine the intricacies of phishing operations without risking unauthorized access to active malicious servers. Lastly, to address ethical concerns surrounding the analysis of phishing kit backdoors, we performed these investigations locally rather than on active phishing servers. While this methodology may not perfectly replicate the exact configurations of live servers, it provided a safe and responsible means of studying these malicious tools.

## 8 Related Work

Previous research has largely overlooked the analysis of header information in phishing threats. Earlier studies have focused on understanding HTTP headers in benign websites and their role in mitigating security attacks.

**Server Information in Web Headers.** Research on security headers has mainly focused on benign websites [11, 24, 30, 36, 57], aiming to understand HTTP response header implementation. Although some studies, such as [11, 24], address security-related headers, their analyses lack depth and are limited to benign sites. Our work stands apart by providing an in-depth analysis of security-related headers used in real-world phishing attacks.

**Measurement of Web Security.** Previous studies have primarily concentrated on benign websites, with works such as [45] examining how developers implement `Content-Security-Policy` and [36] exploring security issues arising from inconsistent HTTP header implementation across different device versions. Unlike these studies, our research focuses on phishing websites, using real-world data to uncover how attackers configure HTTP headers.

**Phishing Ecosystem.** Research efforts such as [38, 39, 41] have explored phishing techniques through controlled experiments, while other studies[3, 26, 32] have investigated existing mitigation strategies and detection mechanisms. These works tend to emphasize the general structure of phishing websites and evasion tactics. In contrast, our study dives deeper into the phishing ecosystem, analyzing configurations through HTTP response headers and phishing kits to better understand attackers' security practices.

**Phishing Kit.** While studies like [51] have focused on analyzing phishing kits to understand their design, and others [40] have examined their use in attacks and current mitigation strategies, our approach aims to identify vulnerabilities within phishing kits and measure how they configure HTTP headers. We also highlight the presence of backdoors in phishing kits, shedding light on how attackers maintain control over compromised sites and exfiltrate sensitive information. Through this analysis, we provide insights into the tactics attackers use to sustain their malicious activities and exploit unsuspecting users.

## 9 Conclusion

Our measurement study provides crucial insights into the security configurations of real-world phishing websites and the vulnerabilities within phishing kits for 31 months. Despite the prevalence of HTTP headers, our findings reveal that they are frequently underutilized or incorrectly configured on phishing websites, highlighting substantial deficiencies in security practices. Our in-depth analysis of 13,344 phishing kits using advanced static analyzers uncovered a high number of vulnerabilities, with almost 90% of the kits exhibiting multiple CWE vulnerabilities.

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

## A    Additional Security Headers

**Deprecated Headers.** Once a header is deprecated, browsers discontinue support for it. This leads to compatibility issues with newer browser versions, as included headers may not function correctly when deprecated headers are used. The two deprecated headers (`Expect-CT` and `X-Frame-Options`) are found in our list of the top 30 most used headers.

**Expect-CT.** The `Expect-CT` header allows websites to opt into reporting and/or enforcing Certificate Transparency (CT) requirements. Although the `Expect-CT` header was deprecated in June 2021 [22], its usage only significantly declined in Sept. 2022, more than a year after it became obsolete, as depicted in Figure 3(h).

**X-Frame-Options.** The CSP includes the `frame-ancestors` directive, which replaces `X-Frame-Options` header. The `X-Frame-Options` header indicates whether a browser should allow a page to render within a `<frame>`, `<iframe>`, `<embed>`, or `<object>`, helping websites prevent clickjacking attacks and ensuring their content isn't embedded within other sites. Although it's used in 5.7% of domains, its usage is rising, as shown in Figure 3(i).

**Deprecated Directives.** Moreover, the `allowall` directive in the `X-Frame-Options` header is not valid. If browsers encounter an invalid directive, it will not be effective, potentially causing the browser to block `<iframe>` elements. In our dataset, `allowall` is the third most used directive, accounting for 21.5% of `X-Frame-Options` usage, despite its ineffectiveness.

**Content-Security-Policy Configuration in HTML.** HTTP headers can also be configured using the `<meta>` tag in HTML, though this practice has been relatively uncommon in the past, as noted in previous research by Roth et al [45]. We investigate further how phishing websites configure headers via the `<meta>` tag and found that 74.4% of the domains (674,661) utilize this tag to set HTTP headers. However, only 0.8% of these domains (5,337) use the `<meta>` tag to set the `Content-Security-Policy` header.

## B    Headers in Phishing Kits

As summarized in Table 4, we analyze the PHP script files included in the phishing kits and find HTTP header configurations in 10.73% scripts (324,071 out of 3,019,018.

**Cache-control.** `Pragma: no-cache` or `Cache-Control: no-store, no-cache, must-revalidate` are used to prevent caching of sensitive pages. This ensures that potential victims constantly receive the latest version of the phishing page, making it easier for attackers to update and modify their phishing kits without the risk of serving outdated content. We find that 3,292 PHP files use the `Pragma: no-cache` header, while 3,001 PHP scripts employ the more comprehensive `Cache-Control` header with multiple directives.

**Redirection.** Redirection is a method where headers like `Location` and `Expires` control the navigation flow within the phishing kit. Expires header contains a specific date and time when the page should be considered expired. If the victim tries to access the same

phishing page again after the specified expiration time, the browser will consider the cached version as expired and send a new request to the server to fetch an updated version of the page.

Our analysis reveals that 7,778 PHP files within the phishing kits utilized relative URL paths in `Location` headers, such as `Location: ../../`, allowing attackers to redirect victims to different pages within the phishing kit. This tactic enables the creation of elaborate phishing campaigns that guide victims through pages designed to harvest sensitive information, increasing the chances of successful attacks. We note that dynamic redirection, exemplified by variables like `location: $dst`, adds an additional layer of complexity to phishing kits. By enabling personalized navigation based on specific conditions or user interactions, attackers can create highly targeted and adaptive phishing experiences. In addition, we discover that 12,283 PHP files employed the `Expires` header to set expiration dates in the past, such as `Expires: Mon, 26 Jul 1997 05:00:00 GMT`. By explicitly telling the browser that the phishing page has already expired, attackers force a reload of the latest version, ensuring that victims interact with the most up-to-date iteration of the phishing kit.

**Content-Security-Policy in Phishing Kits.** `Content-Security-Policy` headers are rarely utilized within the analyzed phishing kit, with only 10 out of the total PHP script files implementing strict CSP rules. The lack of widespread adoption of defensive CSP configurations suggests that the phishing kit's authors have not given much importance to security measures to safeguard their content from detection or analysis by security tools.

Our analysis of 711 PHP script files from various phishing kits reveals that attackers are actively exploiting and manipulating CSP headers to create more permissive and evasive environments for their phishing pages, effectively undermining the security benefits of CSP. One particularly insidious method phishing kits employ is the dynamic generation of CSP headers based on configuration values. As illustrated in Listing 3, attackers can dynamically specify the allowed websites for framing the phishing page using the `frame-ancestors` directive.

```php
public function removeCspHeader(ResponseEvent $event): void{
    if ($this->debug && $event->getRequest()
    ->attributes->get('_remove_csp_headers', false)){
    $event->getResponse()->headers
    ->remove('Content-Security-Policy'); } }
```

**Listing 3: Example of Header Manipulation in PHP.**

**Cloaking.** `Cloaking` is a common method used in phishing kits to serve different content based on the characteristics of the incoming request. Our analysis reveals that 7,367 distinct PHP files within the phishing kits contained conditional HTTP responses, enabling attackers to deliver tailored content based on the requests' properties. By analyzing headers like `User-Agent` and `Referer`, phishing kit authors implement cloaking mechanisms that respond with deceptive error statuses. When a request originates from a suspicious source (*e.g.*, bots, crawlers, detectors), the phishing kit responds with deceptive error statuses like `HTTP/1.0 404 Not Found`. This cloaking behavior aligns with CWE-601, as it involves redirecting victims to malicious pages while showing benign content to security scanners. In addition, phishing kit authors may resort to obfuscation methods to conceal the cloaking logic and

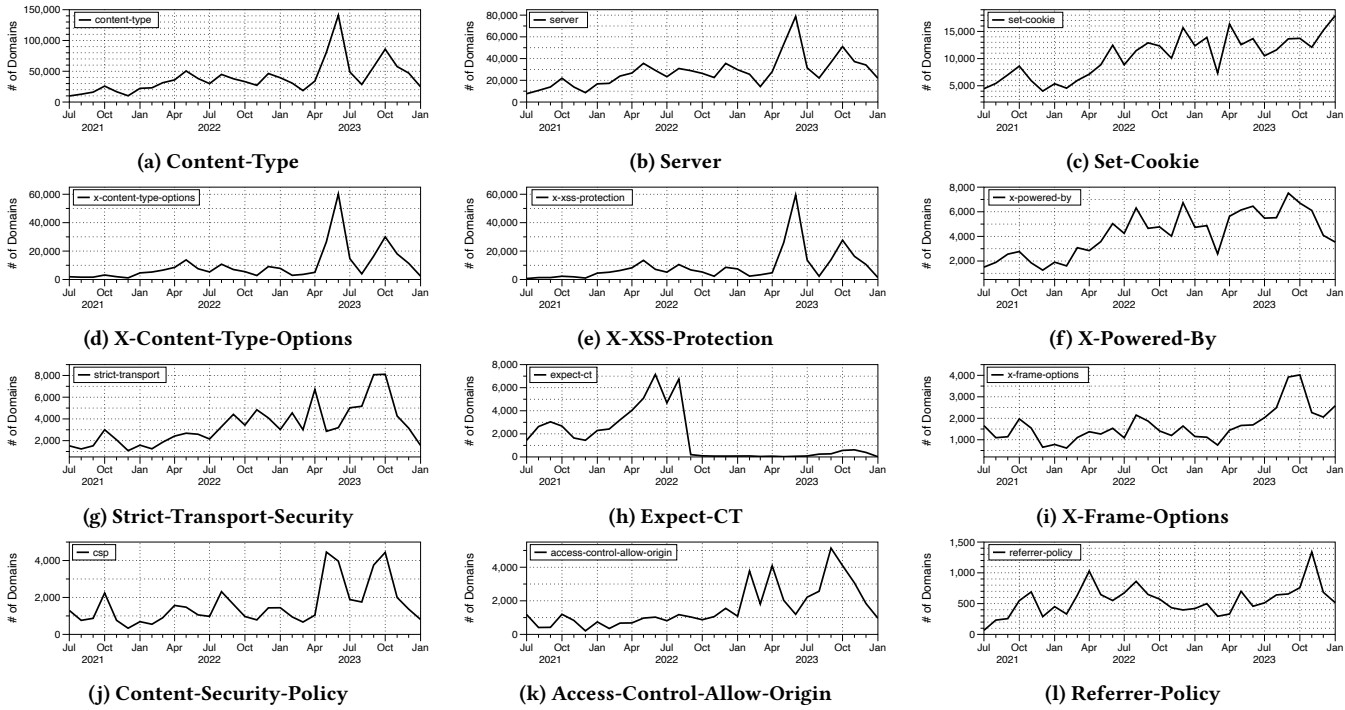

**Figure 3: Top 12 Security-related Headers Usage**

**Table 4: Headers in PHP Phishing Kits.**

| Type | Header | Description |
|---|---|---|
| **Cache -Control** | 'Pragma: no-cache' | Prevents caching of sensitive pages in phishing kits, ensuring the latest version is served to victims. An HTTP/1.0 header is still supported for backward compatibility. |
| | 'no-store, no-cache, must-revalidate, max-age=0' | Combines multiple cache-control directives to prevent caching of the phishing page and ensure that the content is always revalidated. This is crucial in phishing kits to ensure that the victim always sees the latest version of the phishing page and cannot access cached versions. |
| **Redirection** | 'Expires: Mon, 26 Jul 1997 05:00:00 GMT' | Sets an expiration date in the past, effectively telling the browser that the phishing page has already expired and should not be cached. This ensures that the victim always sees the latest version of the phishing page. |
| | "Location: ../../../../" | Redirects the victim to a different URL within the phishing kit. Instructs the browser to navigate the specified relative path, which could lead to another phishing page or a page designed to collect sensitive information. |
| | "location: $dst" | Performs a dynamic redirection to a URL specified by the variable $dst. Phishing kits may use this to redirect victims to different pages based on certain conditions or to navigate them through a series of pages to trick them into providing sensitive information. |
| **Content -Security -Policy** | "frame-ancestors". $cfg->getAllowIframes().";" | Sets the CSP to restrict which sources can embed the phishing page in an iframe. Phishing kits may use this to control where the phishing page can be embedded, potentially limiting the ability of security researchers or anti-phishing tools to analyze the kit. |
| | "script-src 'self'; connect-src 'none'; font-src 'none'; style-src 'self' " | Defines a strict CSP that only allows scripts from the same origin ('self'), disables external connections (connect-src 'none'), disables external fonts (font-src 'none'), and only allows stylesheets from the same origin (style-src 'self'). Phishing kits use this to restrict the resources that can be loaded by the phishing page, making it harder for security tools to detect or analyze the kit. |
| **Cloaking** | "User-Agent: HTTP/1.0 404 Not Found" | Checks the User-Agent header of the incoming request and responds with a 404 Not Found status if the User-Agent doesn't match the expected value. Phishing kits may use this technique to serve different content or responses based on the User-Agent, potentially hiding the phishing page from web crawlers. |
| | "Referer: HTTP/1.0 403 Forbidden" | Checks the Referer header of the incoming request and responds with a 403 Forbidden status if the Referer doesn't match the expected value. Phishing kits may use this to prevent access to the phishing page if the request comes from an unknown or suspicious referrer, making it harder for security researchers to investigate. |

bypass detection. These involve encoding header directives using hex or octal representations.

## C Phishing CPanel and Backdoor in Kits

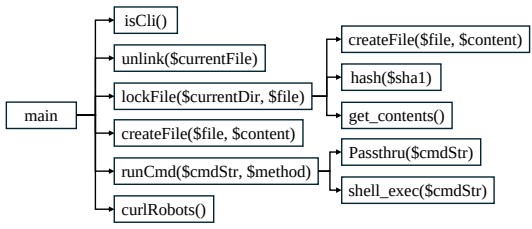

**(a) Reverse Shell Call Tree.**

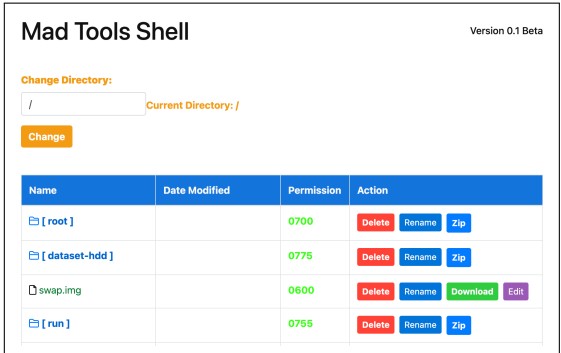

**(b) Example of Phishing Campaign Control Panel.**

**Figure 4: Phishing Control Panel and Backdoor Call Tree.**

Figure 4 presents two key components of a phishing operation: a reverse shell call tree and a phishing campaign control panel. These elements provide insight into the technical infrastructure and user interface employed by threat actors in sophisticated phishing attacks. The reverse shell call tree, shown in Figure 4(a), illustrates the structure of a backdoor program. This program appears designed for remote access and control of a compromised system. The main function serves as the entry point, calling various specialized functions. These functions perform tasks such as checking the execution environment, manipulating files, executing system commands, and potentially interacting with web servers. Notable functions include deleting files (possibly to hide evidence), locking files, creating new files, and executing shell commands through different methods. This structure allows the attacker to maintain access and perform a wide range of actions on the infected system.

Figure 4(b) displays an example of a phishing campaign control panel labeled "Mad Tools Shell". This web-based interface, albeit in a beta version, provides attackers a user-friendly way to manage their phishing infrastructure. The panel allows for directory navigation and file management on the compromised server. It lists files and directories with their permissions and modification dates and offers options to delete, rename, compress, and, in some cases, edit or download files. This interface streamlines the process of managing stolen data and maintaining the phishing site, making it easier for attackers to operate their campaigns efficiently.

**Table 5: Web Shell & System Access Keywords in Kits.**

| Type | Keyword Value |
|---|---|
| Shell | "angel"; "b374k"; "bv7binary"; "c99"; "c100"; "r57"; "webroot"; "kacak"; "symlink"; "h4cker"; "webadmin"; "gazashell"; "locus7shell"; "syrianshell"; "injection"; "cyberwarrior"; "ernebypass"; "g6shell"; "pouyaserver"; "saudishell"; "simattacker"; "sosyeteshell"; "tryagshell"; "uploadshell"; "wsoshell"; "weevely"; "zehir4shell"; "lostdcshell"; "commandshell"; "mailershell"; "cwshell"; "iranshell"; "indishell"; "g6sshell"; "sqlshell"; "simshell"; "tryagshell"; "zehirshell"; "unknown"; "k2ll33d"; "b1n4ry"; |
| System | "pcntl"; "assert"; "passthru"; "shell_exec"; "exec"; "base64_decode"; "edoced_46esab"; "eval"; "system"; "proc_open"; "popen"; "curl_exec"; "php_uname"; "tcpflood"; "udpflood"; "curl_multi_exec"; "parse_ini_file"; "gzinflate"; "show_source"; "phpinfo"; "readfile"; "fclose"; "fopen"; "mkdir"; "gzip"; "python_exec"; "str_rot13"; "chmod"; |

To further understand the components of these backdoors and web shells, Table 5 provides a comprehensive list of keywords commonly associated with backdoors and web shells found in phishing kits. The table is divided into two main categories: "Shell" and "System." The "Shell" category lists various names and types of web shells that attackers might use to maintain unauthorized access to compromised web servers. These include well-known shells like "c99," "r57," and "b374k," and region-specific shells such as "syrianshell" and "indishell."

The "System" category enumerates PHP functions and commands often used in malicious scripts to execute system-level operations. These include functions for executing shell commands (*e.g.* "shell_exec," "exec"), file manipulation ("fopen," "mkdir"), and potentially harmful operations like "eval" that can execute arbitrary PHP code. This table is a valuable reference for security professionals and researchers who want to identify potential backdoors in web applications and phishing kits.

## D PHP Code Analysis of a Phishing Kit

```php
1  <?php
2  require_once('geoplugin.class.php');
3  $geoplugin = new geoPlugin();
4  $geoplugin->locate();
5  $date = gmdate ("Y-n-d");
6  $time = gmdate ("H:i:s");
7  $browser = $_SERVER['HTTP_USER_AGENT'];
8  $message .= "=============+ LOGS +=============\n";
9  $message .= "Email:    ".$_POST['email']."\n";
10 $message .= "Password: ".$_POST['pass']."\n";
11 $message .= "Referer:  ".$_POST['referer']."\n";
12 $message .= "Host:     ".$_POST['host']."\n";
13 $message .= "============= [ip] =============\n";
14 $message .= "IP: {$geoplugin->ip}\n";
15 $message .= "City: {$geoplugin->city}\n";
16 $message .= "Region: {$geoplugin->region}\n";
17 $message .= "Country Name: {$geoplugin->countryName}\n";
18 $message .= "Country Code: {$geoplugin->countryCode}\n";
19 $message .= "User-Agent: ".$browser."\n";
20 $message .= "Date Log  : ".$date."\n";
21 $message .= "Time Log  : ".$time."\n";
22 $domain = "AUTO Logs";
23 $subj = "DOPE LOG !";
24 $from = "From: $domain<west>\n";
25 mail("[AttackerID]@gmail.com, [AttackerID]@yandex.com",
26     $subj,$message,$from,$domain);
27 header("Location: http://www.aliyun.com/");
28 ?>
```

**Listing 4: Example of PHP Used in a Phishing Kit to Exfiltrate Victim Data.**

Listing 4 presents a PHP script commonly found in phishing kits designed to capture and exfiltrate victim data. This code snippet provides valuable insights into the mechanics of data theft and

**Table 6: Top 30 Most Used Headers by Phishing Websites.**

| Rank | Total Usage | | Content Publishing Service | | Self-hosting Server | |
|---|---|---|---|---|---|---|
| | Header | Usage (%) | Header | Usage (%) | Header | Usage (%) |
| 1 | Content-Type | 905,666 (99.9%) | Date | 346,118 (99.9%) | Content-Type | 559,549 (99.8%) |
| 2 | Date | 905,471 (99.9%) | Content-Type | 346,116 (99.9%) | Date | 559,352 (99.8%) |
| 3 | Server | 857,669 (94.6%) | Server | 315,632 (91.1%) | Server | 542,036 (96.7%) |
| 4 | Content-Encoding | 784,233 (86.5%) | Content-Encoding | 313,239 (90.4%) | Connection | 520,838 (92.9%) |
| 5 | Transfer-Encoding | 705,081 (77.8%) | Transfer-Encoding | 290,673 (83.9%) | Content-Encoding | 470,993 (84.0%) |
| 6 | Connection | 633,548 (69.9%) | Cache-Control | 281,289 (81.2%) | Transfer-Encoding | 414,408 (73.9%) |
| 7 | Cache-Control | 586,096 (64.6%) | Etag | 266,892 (77.1%) | Vary | 414,313 (73.9%) |
| 8 | Vary | 494,607 (54.5%) | Alt-Svc | 262,513 (75.8%) | Cache-Control | 304,806 (54.4%) |
| 9 | Expires | 482,550 (53.2%) | Last-Modified | 260,415 (75.2%) | Set-Cookie | 299,240 (53.4%) |
| 10 | Alt-Svc | 475,520 (52.4%) | Expires | 239,165 (69.0%) | Expires | 243,384 (43.4%) |
| 11 | Last-Modified | 349,289 (38.5%) | X-Content-Type-Options | 233,683 (67.5%) | Pragma* | 224,195 (40.0%) |
| 12 | Etag | 338,112 (37.3%) | X-XSS-Protection | 225,737 (65.2%) | Alt-Svc | 213,007 (38.0%) |
| 13 | Set-Cookie | 321,920 (35.5%) | Connection | 112,709 (32.5%) | CF-Ray | 201,163 (35.9%) |
| 14 | X-Content-Type-Options | 300,980 (33.2%) | Vary | 80,293 (23.2%) | CF-Cache-Status | 197,460 (35.2%) |
| 15 | X-XSS-Protection | 280,758 (31.0%) | Content-Length | 55,473 (16.0%) | Report-To | 191,950 (34.2%) |
| 16 | Pragma* | 242,841 (26.8%) | Strict-Transport-Security | 55,010 (15.9%) | NEL | 190,722 (34.0%) |
| 17 | CF-Ray | 228,085 (25.2%) | Accept-Ranges | 41,147 (11.9%) | Keep-Alive | 183,300 (32.7%) |
| 18 | CF-Cache-Status | 217,813 (24.0%) | X-Cache | 30,108 (8.7%) | Content-Length | 145,055 (25.9%) |
| 19 | Report-To | 212,972 (23.5%) | X-Served-By | 27,319 (7.9%) | X-Powered-By | 120,063 (21.4%) |
| 20 | Keep-Alive | 207,735 (22.9%) | X-Cache-Hits | 27,233 (7.9%) | Last-Modified | 88,874 (15.9%) |
| 21 | NEL | 207,342 (22.9%) | X-Timer | 27,228 (7.9%) | Etag | 71,220 (12.7%) |
| 22 | Content-Length | 200,529 (22.1%) | CF-Ray | 26,922 (7.8%) | X-Content-Type-Options | 67,297 (12.0%) |
| 23 | X-Powered-By | 129,992 (14.3%) | Keep-Alive | 24,435 (7.1%) | Accept-Ranges | 64,675 (11.5%) |
| 24 | Accept-Ranges | 105,822 (11.7%) | Set-Cookie | 22,680 (6.5%) | X-XSS-Protection | 55,021 (9.8%) |
| 25 | Strict-Transport-Security | 103,724 (11.4%) | Content-Security-Policy | 21,384 (6.2%) | Strict-Transport-Security | 48,714 (8.7%) |
| 26 | Expect-CT | 51,454 (5.7%) | Access-Control-Allow-Origin | 21,126 (6.1%) | X-Request-ID | 48,090 (8.6%) |
| 27 | X-Frame-Options | 51,280 (5.7%) | Report-To | 21,022 (6.1%) | X-Frame-Options | 45,520 (8.1%) |
| 28 | Content-Security-Policy | 49,130 (5.4%) | CF-Cache-Status | 20,353 (5.9%) | Expect-CT | 41,031 (7.3%) |
| 29 | Access-Control-Allow-Origin | 49,061 (5.4%) | Age | 19,872 (5.7%) | Upgrade | 37,045 (6.6%) |
| 30 | X-Cache | 48,817 (5.4%) | Pragma* | 18,645 (5.4%) | X-Host | 33,783 (6.0%) |
| **Total** | | 906,731 (100%) | | 346,366 (100%) | | 560,538 (100%) |

∗ = Deprecated headers [27]; Insecure HTTP headers are highlighted in Red .

transmission in phishing attacks. The script begins by utilizing the geoPlugin class to gather geographical information based on the victim's IP address. It then collects various data points, including the current date and time and the user's browser information.

A formatted message is constructed, incorporating sensitive information stolen from the victim. This includes the user's email address and password, likely obtained through a deceptive login form. The script also captures referrer information and the host, which can help attackers understand the effectiveness of their phishing campaign. Geographical data obtained from geoPlugin is appended to the message, providing attackers with detailed location information about their victims. This includes the IP address, city, region, country name, and country code. The script uses PHP's mail() function to send the collected data to the attacker's email addresses. Notable security issues include the hardcoding of attacker email addresses directly in the script, making it easier to trace the attack back to its source. Finally, the script redirects the victim to a legitimate website (in this case, Alibaba Cloud), likely in an attempt to avoid suspicion after the data theft has occurred.

