# OpenReview forum: "What’s in Phishers: A Longitudinal Study of Security Configurations in Phishing Websites and Kits"
_ACM.org/TheWebConf/2025/Conference — WWW 2025 Oral_

### Official Review · Reviewer_aFh1 · 2024-11-27

**Novelty:** 4
**Technical Quality:** 3

**Review:**

### Quality
The quality of the paper is good. On the positive side, the authors have put together a really large dataset, which is great. They’ve collected over 13,000 phishing kits and analyzed nearly a million phishing websites, which is no small feat.

### Clarity
The paper is structured in a way that makes it relatively easy to follow, and the authors have done a good job of explaining their methodology and presenting their data in tables and figures. But there’s a bit of a disconnect between what the paper sets out to do and what it actually delivers (see my comments below).

### Originality

First of all, the scope/topic (measuring the security of the phishing kits) of the work appears to be new. Also, the authors have collected a huge dataset, which is valuable in itself.  The findings and techniques not particularly new. Most of what they talk about (e.g., misconfigured HTTP headers, vulnerabilities such as SQL Injection and XSS) has already been well-studied. The idea of using these vulnerabilities to disrupt phishing campaigns could have been an interesting angle, but it’s not explored in a in-depth way.

### Significance

There’s no denying that phishing is a big problem, but this paper doesn’t really move the needle in terms of what we can do about it. How do we take the findings in this paper and use them to stop phishing attacks? The authors don’t answer that question.



### Strengths:
- The authors collected a massive amount of data.
- Some of the findings are interesting.
- The exploration of backdoors in phishing kits adds an interesting layer to the discussion.

### Weaknesses
- The results can be improved by in-depth analysis.
- The reliability of the results/attacks are questionable (which weaken the idea and the overall motivation).
-  There are gaps in the data collection methodology.

### Detailed comments

The results can be improved by in-depth analysis.  For example, the authors highlight that phishing websites rarely implement security headers like Content-Security-Policy compared to benign websites (5.4% versus 75.2%). However, they don’t explain why phishing websites are set up this way. Is it because attackers deliberately avoid these headers, or is it a side effect of outdated tools and kits? What does this finding mean for defense strategies?

The vulnerabilities identified in phishing kits were mostly detected using static analysis tools like Semgrep and Progpilot. While these tools are valuable, they don’t confirm whether the vulnerabilities are actually exploitable in live environments. For instance, the XSS vulnerability described in the verify.php file  is only discussed theoretically. Similarly, the SQL Injection vulnerability in esestandard.zip is not demonstrated against a real database. Why don’t the authors directly test the websites by launching attacks? I understand that launching live attacks might raise ethical concerns, but a key motivation of this paper is to disrupt phishing websites by identifying vulnerabilities and potentially leveraging these insecure configurations to actively exploit phishing kits. If conducting such live attacks is ethically prohibited, does that mean the authors' stated motivation is not valid or realistic? This disconnect between the stated goal and the actions taken in the study needs to be addressed to clarify the paper’s feasibility and ethical considerations.

Now, please tell me, the motivation of exploiting vulnerabilities in phishing kits could help defenders actively disrupt phishing operations, valid or not? The paper repeatedly claims this. However, this idea is presented in an oversimplified manner. In reality, attackers are likely to notice and mitigate attempts to exploit their infrastructure. Moreover, exploiting vulnerabilities on active phishing servers raises serious ethical and legal questions, none of which are addressed in sufficient depth. This makes the paper’s central claim feel impractical and, frankly, contradictory.

Finally, there are gaps in the data collection methodology. The authors  don’t explain how they ensured the kits were authentic and untampered (while the authors have a simple observation, i.e., phishing attackers may leave their kits publicly accessible and downloadable at certain URL paths, it is unclear why it is and how reliable this method is). This lack of transparency raises questions about the validity of the dataset and the generalizability of the findings. I think the authors should conduct FP/FN analysis of the data collection methodology to give us a insights on the result reliability.

**Questions:**

- Could you clarify how reliable the dataset is? Do the observations always hold true in your case?

- Also, can you explain the motivation behind this work: If you haven’t really tested these vulnerabilities, how could you or anyone else use them to actually compromise those websites? And on top of that, wouldn’t attacking phishing websites raise ethical concerns?

**Ethics Review Description:**

Exploiting vulnerabilities (i.e., the idea) on active phishing servers raises serious ethical and legal questions, none of which are addressed in sufficient depth

**Ethics Review Flag:**

Yes

**Reviewer Confidence:**

3: The reviewer is confident but not certain that the evaluation is correct

**Scope:**

4: The work is relevant to the Web and to the track, and is of broad interest to the community

---

### Official Review · Reviewer_qxci · 2024-11-30

**Novelty:** 5
**Technical Quality:** 5

**Review:**

**Summary of paper**

The paper presents a measurement study on the security configurations and vulnerabilities in
phishing websites and associated kits. The paper has collected over 16.7 million distinct phishing URLs over 2 years, and analyzed more than 900k phishing websites. The paper presents a detailed analysis on HTTP headers, vulnerabilities, and misconfigurations, and shows that phishing websites often employ weak security configurations.


**Pros**
+ Overall a well-written paper studying an interesting topic.
+ A large-scale measurement study over 2 years.
+ The paper conducts an analysis of vulnerabilities and misconfigurations on both phishing websites and kits.


**Cons**
- Motivation needs better justification.
- Disconnect between phishing websites and kits.
- Some background information and analysis are missing.


**Comments**

Overall this is a well-written paper. The paper conducts a large-scale measurement study on 900k phishing websites over two years, and collects over 10k phishing kits. The paper shows some interesting findings, demonstrating that phishing websites often lack proper security headers, contain known vulnerabilities, and have misconfigurations.

Some issues. First, the motivation needs better justification. Section 3 is just one paragraph. Phishing detection is mentioned several times throughout the paper. Particularly, by analyzing vulnerabilities in phishing websites and kits, common flaws can be uncovered as indicators for detecting phishing sites but also taking direct action against them. However, how exactly these can help is unclear. For example, how to distinguish phishing websites from poorly designed/implemented websites?

The data source relies on eCX, which is one of the largest phishing attack report repositories containing reliable real-world phishing attack reports. This platform needs more introduction. How does eCX identify phishing websites? Are there false positives？

One missing point is the adoption of HTTPS. How many phishing websites use HTTPS? If most phishing websites simply use HTTP, discussions on vulnerabilities and misconfigurations might be less irrelevant since they are not designed for secure usage.

Section 6 discusses vulnerabilities in phishing kits, which are collected from phishing websites. However, the popularity and impact of these kits are missing. For example, Section 6.2 discusses a backdoor case study. 12.5% of kits have such issues. But how many phishing websites are affected? The discussion on phishing kits can be connected with the websites.

More discussions should be conducted on Table 2, as it takes up half a page.

**Questions:**

See comments

**Reviewer Confidence:**

3: The reviewer is confident but not certain that the evaluation is correct

**Scope:**

4: The work is relevant to the Web and to the track, and is of broad interest to the community

---

### Official Review · Reviewer_QpZu · 2024-11-30

**Novelty:** 5
**Technical Quality:** 6

**Review:**

This paper presents a comprehensive measurement study on the security configurations of phishing websites. By collecting a large dataset of phishing website resources and development kit files, the authors analyze critical security elements, including security-related HTTP headers, screenshot, client-side resources, and vulnerabilities within phishing kits. The findings reveal that phishing websites frequently exhibit underutilized or improperly configured security measures. Moreover, the study highlights the exploitable nature of phishing kits, offering valuable insights into the vulnerabilities of phishing infrastructure.

+ The study spans 31 months,  demonstrating a large-scale data collection effort by developing advanced crawlers for phishing websites and phishing kits. Additionally, it introduces effective methods for refining the collected data to eliminate duplication.
+ The paper identifies critical weaknesses in HTTP headers, with specific attention to misconfigurations in self-hosted phishing websites. It identifies backdoors in phishing kits as well.
+ The paper uncovers critical insights into the weak security configurations of phishing website HTTP headers and the vulnerabilities present in phishing development kits.
+ The authors publicly share their collected dataset, fostering transparency and enabling future research in phishing mitigation.


- The paper lacks an analysis of the underlying reasons behind phishing websites' insecure HTTP headers or misconfigurations compared to benign websites, such as whether these choices stem from outdated development kits, insufficient concern for security, or other factors.
- There is no in-depth discussion of why phishing websites prefer certain HTTP header settings, such as the frequent use of the Set-Cookie header or the low adoption of the SameSite directive.
- The research does not compare security configurations between phishing websites and their corresponding benign websites, missing an opportunity to highlight differences and patterns comprehensively.

**Questions:**

- What are the underlying reasons for phishing websites using insecure HTTP headers or misconfigurations compared to benign websites? Is it due to a lack of concern for security, reliance on outdated development kits, or other fundamental factors? Can the phishing website use the same security configuration as the benign website?
- In the header configurations of phishing websites, can you provide an analysis of the reasoning behind specific settings? For example, why is the Set-Cookie header more commonly used, and what factors contribute to the low adoption of the SameSite directive?
- Could you provide a comparison of the security configurations between phishing websites and their corresponding benign websites? This would help highlight the differences and patterns in security practices, and potentially aid in further research on detecting phishing websites based on security configurations.

**Reviewer Confidence:**

2: The reviewer is willing to defend the evaluation, but it is likely that the reviewer did not understand parts of the paper

**Scope:**

3: The work is somewhat relevant to the Web and to the track, and is of narrow interest to a sub-community

---

### Official Review · Reviewer_qZjb · 2024-12-02

**Novelty:** 4
**Technical Quality:** 5

**Review:**

## Paper Summary

This paper presents a longitudinal study of security weaknesses in phishing websites and associated kits. After analyzing over 906k phishing websites, the authors show that phishing websites often employ weak security configurations, presenting a chance for defenders to take proactive actions to disrupt phishing operations.

## Pros

- **Originality**: The work is highly original in its focus on the security configurations of phishing websites (e.g., HTTP headers). It aims to actively disrupt and neutralize phishing operations rather than just detection. It also explores the dimension of phishing kits, and obtain novel findings.
- **Quality:** The dataset collection process results in a large dataset (16.7M URLs refined to 906,731 unique phishing websites and 13,344 phishing kits). The longitudinal approach over 2.5 years offer a new temporal perspective.

## Cons

However, there are some weaknesses that downplay the contribution and solidity of the paper:

1. **Lack of evaluation or case studies on how the security vulnerabilties can be explored**

The paper provides a comprehensive study on the distributional differences between phishing sites and benign sites, encompassing security headers, misconfiguration, vulnerabilities in CWEs. However, the authors did not attempt to exploit the weaknesses beyond detection. Some case studies should make the claim stronger as the security vulnerabilties in phishing sites can be easily exploited.

For example, for web shell backdoors, how can one automatically identify the vulnerability and devise an disruption to the service accordingly?


2. **Motivation could be better justified**

While the authors claim that those security issues of phishing sites could be attacked for good, I am not fully convinced on the motivation. First, the authors did not provide some justification as how these vulnerabilties can be **easily** exploited. If active disruption is hard, why should one bother to do that rather than detection?

In addition, the work mentions that the lack of security headers in phishing services could be used as features or indicators for phishing detection. It would be better if the authors could further justify this claim via statistical testing, specifying what the features would be, and/or building a simple detector that showcases the feasibility.

3. **Presentation and formatting issues**

- The paper did not follow the format of the submission: Why are the line numbers and ACM reference information removed?
- Presentation: the title for section 5 "HTTP Response Headers" is unclear on what it should be about.
- Missing sentence break for the first line of Section 6.1.1.

**Questions:**

1. What are some key insights on the defender side to detect phishing better?
1. Why do the authors focus on backdoors rather than other malware types?
1. I like the discussion of the "Severity of CWEs" to illustrate how likely these issues could be exploited. What are some similar measure for security-related headers in terms of severity or likelihood of attack?

**Reviewer Confidence:**

3: The reviewer is confident but not certain that the evaluation is correct

**Scope:**

3: The work is somewhat relevant to the Web and to the track, and is of narrow interest to a sub-community

---

### Official Review · Reviewer_y4zd · 2024-12-02

**Novelty:** 7
**Technical Quality:** 6

**Review:**

Thank you for submitting this paper. You analyze phishing pages from APWG's eCX to identify security configurations and vulnerabilities in phishing websites and associated phishing kits.

Pros:
+ Well written
+ Informative
+ Large scale study of phishing sites and associated phishing kits

Cons:
- Ethical considerations were limited to the steps the researchers took to minimize harm, rather than the ethical issues that might arise if their recommendations were implemented

Overall, the paper was well written and informative. I liked the 'takeaway' boxes that helped focus the reader. The study involved at-scale analysis and appears to have been undertaken appropriately.

I appreciate you took steps to ensure you acted ethically, which included not collecting victim information and using a controlled virtual environment to avoid unauthorized access. It would perhaps be useful to the reader to have further details on the legal and ethical issues they should consider if they were to implement your recommendations.

**Questions:**

Please provide an overview of the legal and ethical issues readers may need to consider if they were to implement your recommendations.

**Reviewer Confidence:**

2: The reviewer is willing to defend the evaluation, but it is likely that the reviewer did not understand parts of the paper

**Scope:**

4: The work is relevant to the Web and to the track, and is of broad interest to the community